# A programmed cell division delay preserves genome integrity during natural genetic transformation in *Streptococcus pneumoniae*

Matthieu J. Bergé[1,4], Chryslène Mercy[2], Isabelle Mortier-Barrière[1], Michael S. VanNieuwenhze [3], Yves V. Brun[3], Christophe Grangeasse [2], Patrice Polard[1] & Nathalie Campo [1]

Competence for genetic transformation is a differentiation program during which exogenous DNA is imported into the cell and integrated into the chromosome. In *Streptococcus pneumoniae*, competence develops transiently and synchronously in all cells during exponential phase, and is accompanied by a pause in growth. Here, we reveal that this pause is linked to the cell cycle. At least two parallel pathways impair peptidoglycan synthesis in competent cells. Single-cell analyses demonstrate that ComM, a membrane protein induced during competence, inhibits both initiation of cell division and final constriction of the cytokinetic ring. Competence also interferes with the activity of the serine/threonine kinase StkP, the central regulator of pneumococcal cell division. We further present evidence that the ComM-mediated delay in division preserves genomic integrity during transformation. We propose that cell division arrest is programmed in competent pneumococcal cells to ensure that transformation is complete before resumption of cell division, to provide this pathogen with the maximum potential for genetic diversity and adaptation.

[1] Laboratoire de Microbiologie et Génétique Moléculaires, Centre de Biologie Intégrative (CBI), Centre National de la Recherche Scientifique (CNRS), Université de Toulouse, UPS, 31062 Toulouse, France. [2] Microbiologie Moléculaire et Biochimie Structurale (MMSB), Université Lyon 1, CNRS, UMR5086, 69007 Lyon, France. [3] Departments of Biology and Chemistry, Indiana University, Bloomington, IN 47405, USA. [4] Present address: Department of Microbiology and Molecular Medicine, Institute of Genetics and Genomics in Geneva, Faculty of Medicine, University of Geneva, CH-1211 Geneva, Switzerland. Correspondence and requests for materials should be addressed to P.P. (email: patrice.polard@ibcg.biotoul.fr) or to N.C. (email: nathalie.campo@ibcg.biotoul.fr)

In response to DNA damage, both prokaryotic and eukaryotic cells maintain the integrity of their genomes by activating stress responses in coordination with the cell cycle[1,2]. In the majority of bacterial species, most responses to DNA lesions and replication fork failures are ultimately controlled by the SOS regulatory pathway[2,3]. The SOS response relies on RecA-dependent self-cleavage of the master transcriptional repressor LexA and the ensuing derepression of a defined set of genes, including genes involved in DNA repair. Induction of the SOS response also causes cell cycle arrest by blocking cell division, presumably to allow more time for repair and to ensure proper chromosome replication and segregation to daughter cells[4–9]. The SOS pathway is not universal, and many bacteria lacking a LexA homolog have evolved alternative DNA damage responses[2]. Among these bacteria, the human pathogen *Streptococcus pneumoniae* (the pneumococcus) has been proposed to use competence for genetic transformation as a general response to stress[10–12]. Natural transformation is a programmed mechanism of horizontal gene transfer that is widely distributed in bacteria. It has become increasingly evident over the past two decades that

this process contributes to the remarkable plasticity of the pneumococcus and plays a central role in its adaptation to host defenses[13].

Integration of imported DNA into the genome by homologous recombination during transformation requires that cells enter a transient differentiated state termed competence. In pneumococcus cultures, competence is induced during exponential growth. It propagates into the entire population and lasts for a period of about 25 min, shorter than the generation time of a single cell[14]. Pneumococcal competence relies on a secreted peptide pheromone, the competence-stimulating peptide (CSP) that triggers a rapid, global shift in gene transcription and protein synthesis, with three temporally distinct expression profiles: early, late, and delayed[15–17]. The early competence (*com*) set consists of about 20 genes activated by the ComE protein. Among them, the *comX* gene encodes the competence-specific alternative sigma factor ComX[18] required for expression of the late *com* genes. The late *com* group comprises more than 60 genes, including those that encode proteins involved in DNA uptake and the processing of internalized DNA. Ultimately, development of the competence

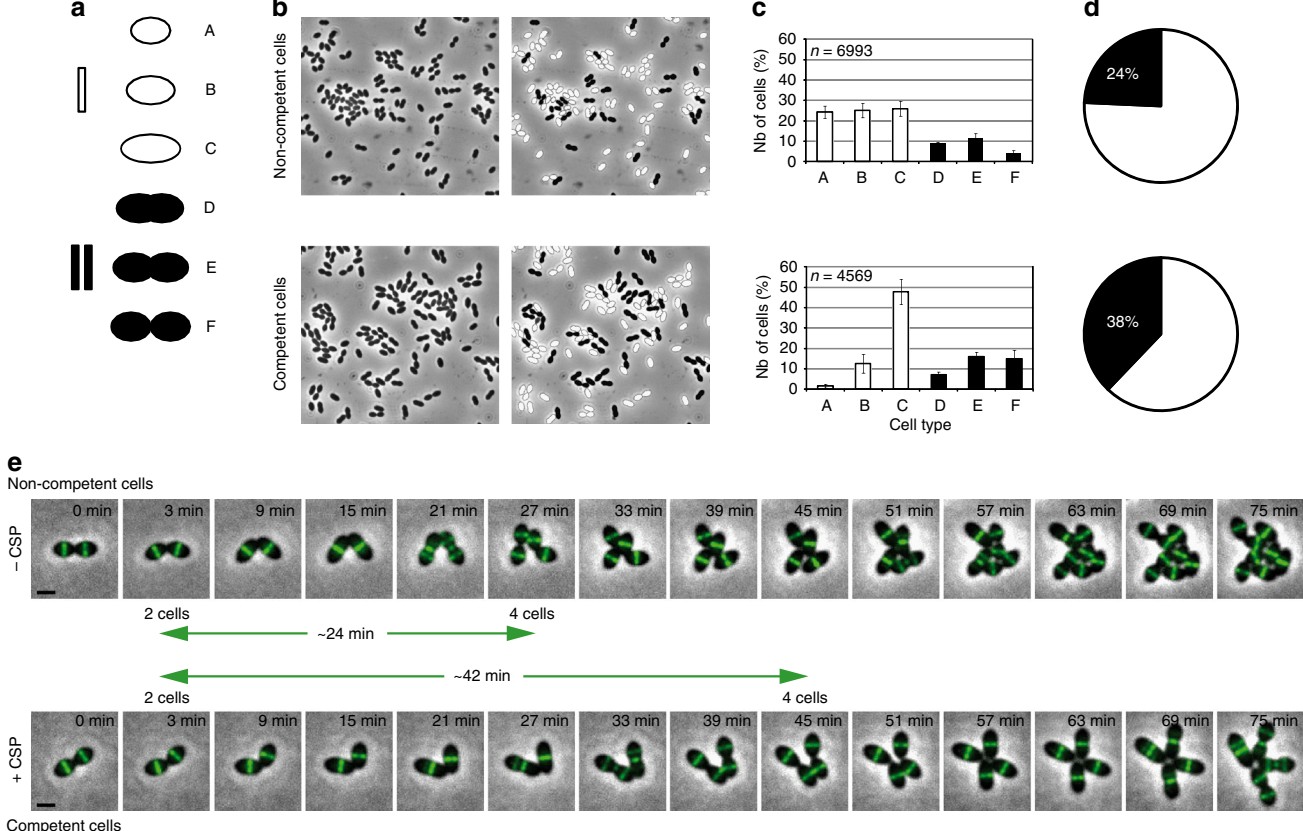

**Fig. 1** Initiation of constriction of the cytokinetic ring and completion of cell division are delayed in competent cells. "Wild type" cells (strain R3956) were incubated at 37 °C with or without CSP for 30 min. **a** Schematic representation of pneumococcal cells classified into two groups and six different classes according to the progression in their cell cycle. **b** Representative fields of non-competent (upper panels) and competent (lower panels) cultures are shown. Left panels: phase contrast images; right panels: same images with cells from group I and group II false-colored white and black, respectively. **c** Histograms quantifying the percent of cells of non-competent (upper) and competent (lower) cultures in the cell-type categories presented in **a**. Note that in non-competent cultures, the proportion of cells in categories A, B and C is similar and that cells from category F represent a minority. Values and standard deviations are based on data from six independent experiments. *n* number of cells analyzed. **d** Pie chart representations of results shown in **c**. The distribution of non-dividing (white) and dividing (black) cells in non-competent (upper chart) or competent (lower chart) cultures is shown. **e** Still images from fluorescence time-lapse microscopy of R3702 cells producing a functional FtsZ-GFP fusion at 37 °C. Cells were induced (+CSP) or not (−CSP) to develop competence by the addition of synthetic CSP and spotted on a microscope slide containing a pad of agarose and C + Y growth medium. Average doubling times of 30 ± 4 min and 42 ± 5 min were measured for a total of 137 non-competent cells and 117 competent cells, respectively, collected over three independent experiments. Overlays between phase contrast (gray) and GFP (green) are shown. The time-lapse starting point corresponds to 5 min of CSP induction. Scale bar, 1 μm

state leads to the induction of roughly 100 genes, of which only 22 are necessary for the transformation process itself[15,17]. Among the other CSP-regulated genes, those contributing to genome plasticity include the late *com* genes *cbpD*, *cibA*, *cibB*, and *lytA*. These genes encode lytic enzymes involved in a killing mechanism, termed fratricide that can be used by competent cells to acquire DNA from non-competent pneumococci[19,20]. Two immunity factors protect competent cells from this process, CibC, which provides protection against the putative two-peptide bacteriocin CibAB[19], and the early competence protein, ComM, which confers immunity to the murein hydrolase CbpD[20].

Stressful environmental conditions, such as exposure to DNA-damaging agents and antibiotics, can induce competence[10,12,21,22]. It is noteworthy that the competence regulon contains genes both for recombination enzymes essential for transformation that can also be used in repair pathways, and for factors such as heat-shock proteins for general response to stress[15,17]. In view of the parallel between competence and stress response, CSP appears to act as an alarmone in this context[10,11]. Notably competence induction has been seen to result in a distinct reduction in growth rate of the cell population[17,20,23]. The cause of this growth rate reduction and its biological relevance are obscure.

Here we explore connections between competence and the cell cycle in *S. pneumoniae*. We show that cell division is delayed in competent cells, and we present evidence that the biological significance of this delay is to contribute to the preservation of genomic integrity during transformation.

## Results

**The cell cycle is altered upon competence induction.** Under optimal conditions, all cells in pneumococcal cultures are able to develop competence (see Supplementary Note 1 and Supplementary Fig. 1)[21,24]. In addition, several reports support the notion that pneumococcal cells developing competence undergo a period of growth inhibition[17,23,25]. Since bacterial cells often have different shapes or sizes depending on their growth rate[26,27], we reasoned that if competent cells are physiologically distinct from non-competent cells, morphological differences between the two populations should be detectable. Accordingly, we compared CSP-induced and non-induced cultures microscopically, by measuring morphological parameters of individual cells. To do this, we developed a standardized method based on the Integrated Morphometric Analysis tool from Metamorph that allowed us to separate pneumococcal cells into six different classes according to progress through the cell cycle (see "Methods"; Fig. 1a and Supplementary Fig. 2). For simplicity, these classes were ordered into two larger groups: cells without constriction (group I), and cells showing constriction at the septal plane (group II).

We first examined a population of growing wild-type non-competent cells. Bacterial cells in culture are asynchronous, with different stages of the cell cycle represented in proportion to their relative duration. *S. pneumoniae* is a rugby ball-shaped bacterium that divides perpendicularly to the long axis of the cell. Its characteristic oval shape is achieved through a combination of peripheral and septal cell wall syntheses, both limited at the site of cell division. Newborn cells (classified as category A in Fig. 1a) arise with most components of the cell division machinery already in place[28]. These cells undergo a short period of cell elongation associated to peripheral cell wall synthesis (Fig. 1a, categories B and C), before cell division[29]. Cells of category C are also called "pre-divisional cells." A distinctive feature of the cell division process in the pneumococcus is that cell constriction is concomitant with septal cell wall synthesis and cell elongation[28,29]. These combined activities, all occurring at the division

site, ensure the formation of the new almost hemispherical halves of the two future daughter cells. Once the division process (cytokinesis) is complete, the two newborn cells remain linked by a peptidoglycan bridge resulting in a cell pair. Figure 1b, top row, shows a representative field of non-competent cells (strain R3956, Supplementary Table 1). Physical parameters were automatically measured for a total of 6993 cells collected over six independent experiments, and the cells were subsequently classified according to their morphological type (Fig. 1a). The proportion of cells in the population displaying each cell type is diagrammed in Fig. 1c. Cells without a constriction (group I) accounted for 76% of the population, while 24% of the cells were dividing (group II) (Fig. 1c, d). This is in agreement with previous results observed for asynchronous cultures of *S. pneumoniae*[29,30]. In contrast, the proportion of dividing cells gradually increased over time in competent cultures reaching a maximum of 38%, 30–40 min after CSP induction (Fig. 1c, d). As these cultures eventually recovered a cell-type distribution similar to that measured in non-competent populations over the next 60–90 min (Supplementary Fig. 3), these results suggest that the constriction process undergoes a deceleration or transient block upon CSP induction. Furthermore, closer examination of cell shape parameters in cultures developing competence reveals a progressive accumulation of pre-divisional cells, corresponding to the C cell-type category (Fig. 1a, c and Supplementary Fig. 3). Similar analysis of dividing cells from group II shows an increase of the number of cells near completion of cell constriction (category F cell type, Fig. 1c). Conversely, newborn cells (category A cell type, Fig. 1a) decrease in the population (Fig. 1c and Supplementary Fig. 3). Finally, plotting the cell dimensions revealed that, while the cell width remains fairly constant in all cell categories regardless of their competent or non-competent state, the cell length increases throughout the cell cycle and this increase is even higher during division of competent cells (Supplementary Fig. 2). Together, this analysis demonstrates that alteration of cell cycle progression in competent cells results from a transient inhibition of two key steps in the division process: initiation and completion of cell constriction. We reached an analogous conclusion when we compared the cell-type distribution of non-competent and competent cultures of the encapsulated strain G54[31] (Supplementary Fig. 2), suggesting that a delay in the cell division process is a general feature of competent pneumococci.

To further examine the CSP-dependent cell division defect, we monitored the doubling time of single cells in competent and non-competent cultures using time-lapse microscopy (Fig. 1e). For this, we used a strain harboring a functional FtsZ-GFP fusion as a marker for cell division. Image analysis clearly shows that cell division is delayed in cells developing competence (Fig. 1e). In our experimental conditions, the cell number doubling time is about 30 min in non-competent cells but increased to more than 40 min after CSP induction. Moreover, measuring the doubling time over several generations indicates that the cell division delay only takes place during the first division event following CSP addition. We conclude from these observations that competent cells undergo a CSP-dependent constriction delay during the division cycle following competence induction.

**Cell division delay does not depend on late *com* gene expression.** We next sought to identify the CSP-dependent gene(s) responsible for delaying cell division. To further test whether the division delay is dependent on the product of one or several late *com* genes, we analyzed the cell distribution of a *comX* mutant strain (strain R2002) with or without CSP induction. In this strain, only early *com* genes are expressed in response to CSP. Figure 2 shows that, in the absence of late *com* genes, cells

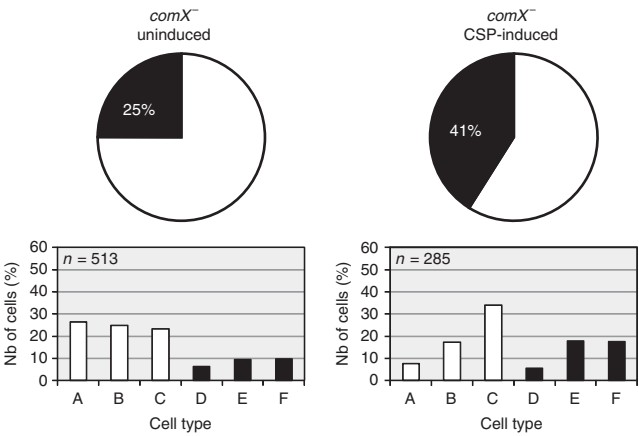

**Fig. 2** Late *com* genes are not required for the cell division delay observed in competent cells. *comX⁻* cells (strain R2002) were CSP-induced to develop the competence program, incubated at 37 °C for 30 min and analyzed by phase contrast microscopy. Pie charts show the distribution of non-dividing (white) and dividing (black) cells in non-competent cultures (indicated as "uninduced", left) or CSP-induced cultures (right). The histogram representations indicate the percentage of cells in the cell-type categories presented in Fig. 1a. *n* number of cells analyzed

developing the competence program still exhibit a constriction defect, indicating that an early *com* gene product is responsible for the observed cell division delay. Importantly, and consistent with the observation that induced transcription of early competence genes persists much longer in a *comX*-deficient strain[23,32], the constriction defect in the *comX* mutant cells persisted more than 90 min after CSP induction (Supplementary Fig. 4). At that time, when wild-type cells have recovered a cell-type distribution resembling non-induced cultures, *comX* mutant cells displayed an inflated and elongated morphology, with some cells harboring multiple aberrant constrictions. These observations imply that a product of the ComE-dependent genes is responsible for interfering with the constriction process in cells developing competence. The transient nature of this defect could be due to the shut-off of early *com* gene expression and/or to the induction of late *com* gene(s) that antagonize this activity.

**The cell division delay depends on the early *com* gene *comM*.** There are about 20 early *com* genes in the *S. pneumoniae* genome[15,17]. Eight of these, *comAB*, *comCDE*, *comX1-comX2*, and *comW*, are necessary for transformation and encode products involved in the competence regulatory cascade. Among the remaining genes, we first selected *lytR* as a potential septation inhibitor. Indeed, *lytR* deletion mutants carrying an unknown suppressor mutation display several septa distributed over multiple sites within the same cell, suggesting that the LytR protein plays a role in controlling septum formation and proper septum placement[33]. *lytR* is the last gene of the largest early *com* operon, which consists of four open reading frames (*comM*-*spr1761*-*spr1760*-*lytR*). *comM* encodes the fratricide immunity protein[20]. *spr1761* and *spr1760* have unknown functions. *spr1761*, *spr1760*, and *lytR* are essential genes and are expressed at a basal level owing to an extended −10 promoter located upstream of *spr1761*[20,33]. Transcription analyses indicate that during competence induction, expression of these genes increases four- to eightfold upon binding of ComE to the *comM* promoter[15,17] (Supplementary Fig. 5a).

We reasoned that if LytR, which is expressed at basal level in non-competent cells, is involved in the cell division delay observed in competent cells, its increased production during

competence development might serve to fulfill this function. To test this possibility, we inserted the strong transcription terminators of *Escherichia coli rrnB*, T1T2, immediately upstream of the extended −10 promoter so as to virtually eliminate ComE-dependent transcription beyond *comM* (Fig. 3 and Supplementary Fig. 5a, see "Methods" section). The resulting strain, R3048, is referred to hereafter as "*comM⁺ lytRⁱⁿᵈ⁻*." In addition, a mutant strain (R3049, "*ΔcomM lytRⁱⁿᵈ⁻*") carrying a deletion of the entire *comM* gene, including its ComE-dependent promoter, was constructed. In this strain, the *spr1761-spr1760-lytR* operon is still expressed at basal level but cannot be induced during competence. We investigated the cell-type distribution of these strains in cultures of competent and non-competent cells. In parallel, *ssbB::luc* expression was monitored as a control for competence development (see Supplementary Note 1 and Supplementary Fig. 5b). As expected, microscopical analysis indicated that the cell distribution of the two mutant strains was indistinguishable from that of wild-type growing in non-competent liquid culture. After CSP induction, pre-divisional and dividing cells increased as a fraction of the total population in competent wild-type and R3048 (*comM⁺ lytRⁱⁿᵈ⁻*) cultures (Fig. 3). We conclude from this experiment that ComE-dependent overexpression of the *lytR* operon is not required for competence-specific cell division delay and that at least one other early *com* gene is required for this function.

On the other hand, examination of the R3049 (*ΔcomM lytRⁱⁿᵈ⁻*) population indicated that the proportion of each cell type remained unchanged after CSP induction (Fig. 3). Although cells carrying the *ΔcomM lytRⁱⁿᵈ⁻* mutation were fully able to develop competence, as established by measurement of *ssbB::luc* expression (Supplementary Fig. 5b) and by detection of fluorescence from Pₓ-*gfp* expression (Supplementary Fig. 5c), they showed no constriction-defective phenotype. This result suggested that ComM either alone or in combination with proteins encoded by the *spr1761-spr1760-lytR* operon is responsible for inhibiting the constriction process during competence development. To discriminate between these possibilities, we took advantage of a *mariner* insertion mutant of *comM*, R1887[20]. It is known that this mutation inactivates the *comM* gene but does not suppress CSP-dependent induction of the *spr1761-spr1760-lytR* operon in competent cells[20] (Supplementary Fig. 5a). Cells were induced to develop competence and visualized by phase contrast microscopy. Figure 3 (bottom row) presents the proportion of cells in each cell category 30 min after CSP induction. The results were typical for a fast growing asynchronous population of cells, similar to those of non-competent cultures, suggesting that in the absence of ComM and even when LytR is overproduced, cell division proceeds normally in competent cells. Altogether, these data indicate that ComM is required to regulate the constriction process during competence development.

Finally, the finding that ComM, known as the fratricide immunity protein[20], was required to inhibit cell division in competent cells prompted us to test CbpD. CbpD is a murein hydrolase identified as the key effector of the fratricide mechanism and as responsible for the killing of a fraction of the *comM⁻* population[34]. After competence induction, the cell-type distribution of a *cbpD* mutant (R1720) appeared indistinguishable from wild type (Supplementary Fig. 6), demonstrating that the hydrolase is not involved in the competence cell division delay. Importantly, a *cbpD comM* double mutant (R1886) behaved like a single *comM* mutant, with no cell division delay in competent cultures (Supplementary Fig. 6). This result implies that the low proportion of dividing cells in competent *comM⁻* cultures is not due to CbpD that could have specifically targeted this cell type in the absence of the major fratricide immunity

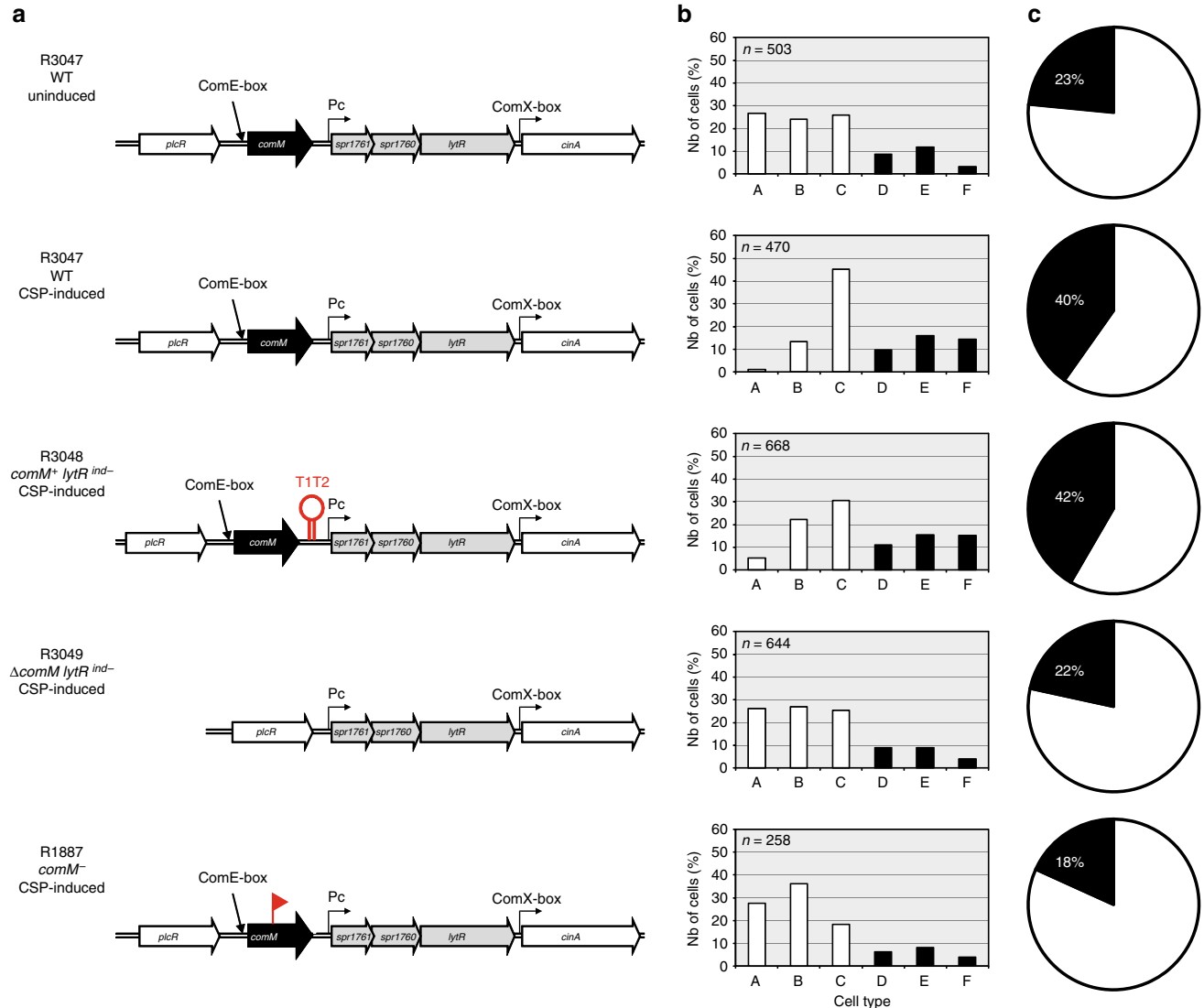

**Fig. 3** ComM is required to delay cell division in competent cells. **a** Diagram of the *comM-spr1761-spr1760-lytR* chromosomal region. The ComE-binding site (*ComE-box*, upstream of *comM*), the extended −10 σ^A-dependent promoter (*Pc*, upstream *spr1761*) and the σ^x-dependent promoter (*ComX-box*, in front of *cinA*) are indicated. The T1T2 transcription terminator (red hairpin) inserted downstream of *comM* and upstream of Pc in strain R3048 (*comM+ lytR^ind−*) and the position of the mariner insertion (red flag) in the *comM* gene of strain R1887 (*comM−*) are also shown. For details of strain constructions, see Supplementary Methods. Cells were induced to develop competence (indicated as "CSP-induced") or not ("uninduced"), incubated at 37 °C for 30 min and analyzed by phase contrast microscopy. **b** Histograms quantifying the percentage of cells in the cell-type categories presented in Fig. 1a. *n* number of cells analyzed. The data are from a single representative of three biological experiments. **c** Pie chart representations of results shown in **b**. The distribution of non-dividing (white) and dividing (black) cells is shown. Strains used: "wild-type" (WT, R3047), *comM+ lytR^ind−* (R3048), Δ*comM* (R3049), *comM−* (R1887)

factor[34], and further confirms the requirement for ComM in the competence-dependent cell constriction delay.

**ComM alone is sufficient to affect the cell dvision process**. To determine whether ComM is the only competence protein required for the cell division delay, we engineered a strain producing this protein in growing non-competent cells. Assuming that the presence of such a protein might impair growth and to avoid selecting for potential suppressors, the *comM* open reading frame was placed at an ectopic locus under the control of a BIP-inducible promoter (see "Methods" section). Bacteriocin inducible peptide (BIP) is a peptide pheromone that activates a pathway leading to bacteriocin production[35,36]. The resulting strain, R3957 (*comM+* P_BIP:*comM*), is therefore able to synthesize the ComM

protein in competent or non-competent cells upon either CSP or BIP induction, respectively. Importantly, the BIP pheromone is completely inactive with respect to competence induction[37]. Cells grown to early exponential phase were divided into three equal cultures then incubated for 30 min with CSP, BIP, or without peptide (Fig. 4a). Cultures containing CSP or BIP both exhibited a higher proportion of dividing cells (37 and 34%, respectively), relative to the non-induced culture (19%). As a control, we used a strain expressing luciferase under the control of the BIP-inducible promoter (R2524, *comM+* P_BIP:*luc*)[38]. In cultures of this strain, the proportion of dividing cells increased after competence induction but remained unchanged upon BIP addition (Fig. 4a), indicating that BIP induction itself has no effect on the cell division process. These results demonstrate that ComM alone can delay cell division, even without the associated development of

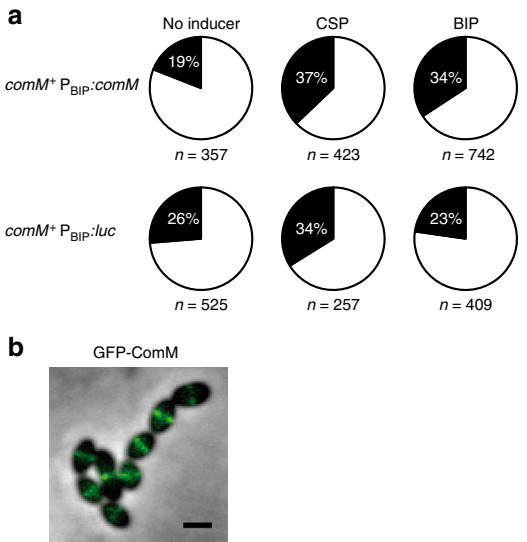

**Fig. 4** ComM is a cell division inhibitor. **a** ComM is sufficient to delay cell constriction in non-competent cells. Pie charts indicate the percentage of non-dividing (white) and dividing (black) cells in cultures of strain R3957 (*comM*⁺ P$_{BIP}$:*comM*) harboring the endogenous copy of *comM* under the control of CSP induction during competence and a second copy expressed independently of competence, under the control of BIP induction; and strain R2524 (*comM*⁺ P$_{BIP}$:*luc*) containing the gene encoding luciferase (*luc*) under the control of BIP induction. Cells were grown in C + Y medium to OD$_{550\ nm}$ 0.08 and induced with or without CSP or BIP as indicated, incubated at 37 ° C for 30 min before phase contrast microscopy analysis. *n* number of cells analyzed. The data are from a single representative of four biological experiments. **b** ComM localizes at midcell. Cells containing a *gfp-comM* translational fusion at the endogenous chromosomal locus (strain R3983) were analyzed by fluorescence microscopy 15 min after CSP addition. Overlays between phase contrast (gray) and GFP (green) are shown. Scale bar, 1 μm. The data are representative of three biological replicates

competence. They are consistent with a recent report showing that ComM overexpression leads to growth inhibition and morphological abnormalities[39]. Finally, localization microscopy analysis of a GFP-ComM fusion indicates that the protein concentrates at midcell, consistent with a role in cell division inhibition (Fig. 4b).

**Competence hinders peptidoglycan synthesis and StkP activity**. We can deduce from Fig. 1e that competence does not delay cell division by disassembling the FtsZ ring. The FtsZ-GFP signal was detected in all competent cells throughout the cell cycle and the localization pattern was undistinguishable from that of non-competent cells (Fig. 1e and Supplementary Fig. 7). The tubulin-like FtsZ protein assembles into a cytokinetic ring at the division site and sequentially recruits a series of cell division proteins to form the divisome[40]. To test whether competence affects the recruitment of other components of the cell division machinery, we analyzed the localization of functional fusions of GFP to FtsW and Pbp2x (FtsI). These proteins, required for the synthesis of septal cell wall, are two of the last components recruited during divisome assembly[28]. Both GFP-Pbp2x and FtsW-GFP localized to the cell division site in competent and non-competent cells (Supplementary Fig. 7). These observations are consistent with the idea that competence development has no effect on the assembly or disassembly of the cell division machinery. We hypothesized instead that the cell division delay occurring in competent cells could result from a transient inactivation or deceleration of the constriction process.

As mentioned above, cell constriction occurs at midcell concomitantly with peripheral and septal peptidoglycan syntheses in pneumococcal cells. In an attempt to assess cell constriction, we monitored peptidoglycan biosynthesis using HADA, a fluorescent D-amino acid probe (FDAA)[41]. In *S. pneumoniae*, growing cells exhibit minimal peptidoglycan turnover and it was shown that FDAA labeling patterns indicate the location of PBP transpeptidase activity, with a distinct demarcation between regions of old peptidoglycan and regions of newly inserted peptidoglycan[42]. Cells were induced to develop competence by addition of CSP for 10 min, incubated with HADA for another 15 min, washed and finally allowed to grow 10–15 min on a microscope slide covered with a pad of nutrient-agarose before imaging. In non-competent cultures, 70% of the cells exhibited a polar hemispherical staining, indicating that the majority of the cells had undergone cell division during incubation with HADA (Fig. 5a, b and Supplementary Fig. 8a). By contrast and consistent with an inhibition of cell division, only 2.6% of the cells harbored this pattern in competent cultures. In this population, we also observed a large proportion of cells with a single thin fluorescent band at midcell (42%). This pattern, which was rarely seen in non-competent cultures (<1% of the cells), suggests that these cells did not grow after HADA removal and that the rate of peptidoglycan synthesis was reduced during competence. This hypothesis is supported by the observation that single band staining eventually gave rise to two discrete bands when the cells were allowed to grow for another 20 min in the microscope chamber (Supplementary Fig. 8b). Another possibility is that the activity of enzymes involved in peptidoglycan turnover, such as carboxypeptisases and endopeptidases, increases during competence and alters the labeling pattern. Interestingly, competent cultures of cells lacking ComM and non-competent cultures of cells expressing ectopic ComM exhibited an intermediate phenotype between non-competent and competent wild-type cultures with 42% of cells presenting a polar staining and 22% a thin fluorescent band at midcell. Moreover, expressing ectopic ComM in non-competent cultures reduced the proportion of cells stained at the poles (40% vs. 70% in non-competent R2524 control cultures, Supplementary Fig. 8a). These results indicate that peptidoglycan synthesis is negatively affected by the presence of ComM. They also suggest that other competence factors could interfere with peptidoglycan synthesis.

In *S. pneumoniae*, the eukaryote-like serine/threonine kinase StkP and its cognate phosphatase play a pivotal role in coordinating cell division and peptidoglycan synthesis through phosphorylation and dephosphorylation of several key proteins[43]. Hence, a catalytically inactive mutant StkP becomes elongated, as peripheral cell wall synthesis exceeds septal cell wall synthesis[44,45]. Notably, the absence of StkP affects competence induction and results in decreased transformation efficiency[46–48]. To investigate whether StkP is involved in the cell constriction delay observed during competence, we compared the phosphorylation profiles of cells in competent and non-competent cultures. Anti-phosphothreonine western blot analysis revealed that the phosphorylation state of at least one StkP substrate, DivIVA, was slightly but reproducibly reduced 10 min after competence induction (Fig. 5c and Supplementary Fig. 8c, d). A reduction in the phosphorylation level of other substrates appeared in some experiments but this variability was not significant on average. DivIVA is a cell division protein that favors synthesis of the cross wall in its phosphorylated form but stimulates cell elongation when dephosphorylated[49]. The fact that the phosphorylation level of DivIVA is only slightly decreased during competence could be the result of both the transient and the asynchronous characteristics of the competent population. A similar decrease occurred in competent cells lacking ComM, suggesting that ComM is not

required for the reduction of the phosphorylation level of DivIVA (Supplementary Fig. 8c, d). However, to strengthen this conclusion, we examined the phosphorylation profiles of *comX*

mutant cultures 40 min after CSP induction. In the absence of ComX, the shut-off of competence is impaired and the expression of early competence genes including *comM* is maintained for a

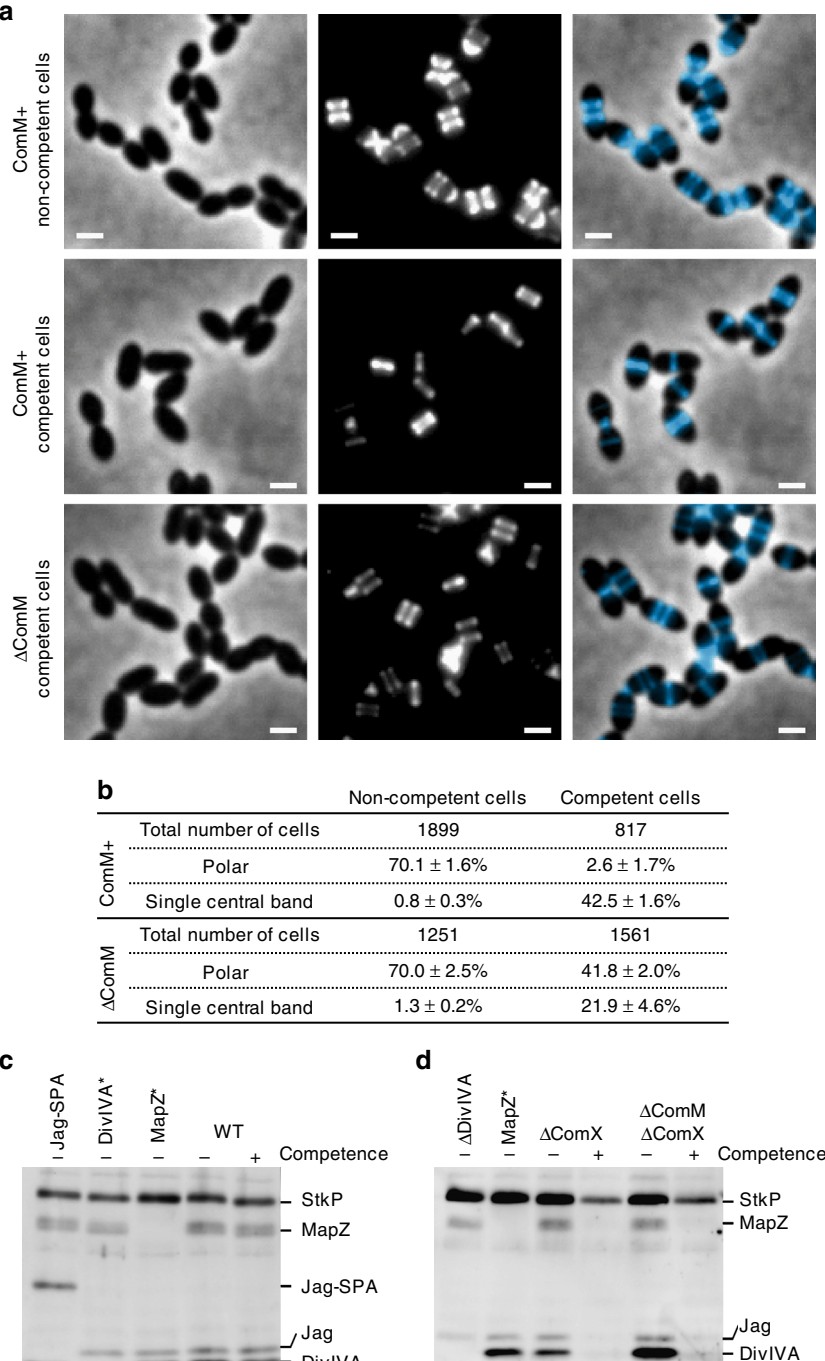

| | | Non-competent cells | Competent cells |
|---|---|---|---|
| **ComM+** | Total number of cells | 1899 | 817 |
| | Polar | 70.1 ± 1.6% | 2.6 ± 1.7% |
| | Single central band | 0.8 ± 0.3% | 42.5 ± 1.6% |
| **ΔComM** | Total number of cells | 1251 | 1561 |
| | Polar | 70.0 ± 2.5% | 41.8 ± 2.0% |
| | Single central band | 1.3 ± 0.2% | 21.9 ± 4.6% |

**Fig. 5** Competence interferes with peptidoglycan synthesis and StkP kinase activity. **a** Localization of peptidoglycan synthesis in the presence (ComM + ) or absence (ΔComM) of ComM. Competent and non-competent cells were incubated for 15 min with the blue fluorescent derivative of D-alanine HADA. Phase contrast (left), fluorescent (middle), and false-colored overlay images (phase contrast, gray; HADA, blue) are shown. Scale bars, 1 μm. Images are representative of seven biological replicates. **b** Fraction of cells harboring polar HADA staining and cells with a single thin HADA band at midcell in ComM+ (R3966) and ΔComM (R3967) cultures. The total number of cells analyzed is indicated. Values are indicated with s.d. for three independent experiments. **c**, **d** Western blot of cell lysates probed with anti-phosphothreonine antibodies. Samples were prepared from competent (+) or non competent (−) cultures. Control strains harboring a deletion of the *divIVA* gene (R4134), a Jag-SPA fusion (R4143), and phosphoablative mutants of DivIVA (DivIVA*, R4144) and MapZ (MapZ*, *mapZ-2TA*) were used to distinguish the phosphorylation signals for StkP, MapZ, Jag, and DivIVA. **c** Strain used: WT (R1501). Data are representative of seven biological replicates and three technical replicates. Full blot is shown in Supplementary Fig. 12a. **d** Strains used: ΔComX (R2002), ΔComX ΔComM (R2132). Data are representative of two biological replicates and one technical replicate. Full blot is shown in Supplementary Fig. 12b

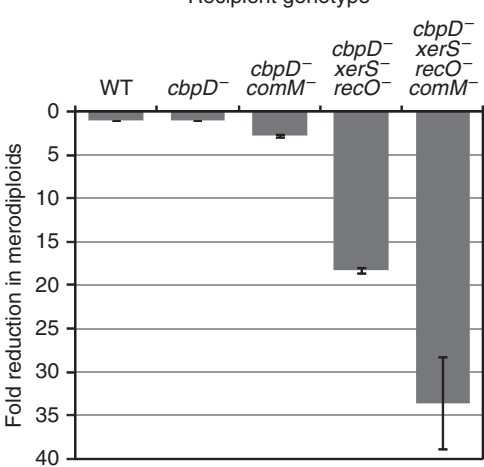

**Fig. 6** ComM facilitates the resolution of chromosome dimers generated during transformation. Reduction in merodiploid frequency in *comM⁻* strains compared with that in wild-type and *cbpD^C75A* cells. Comparison of transformation efficiency using as donor a mixture of PCR fragments that promote merodiploid formation and enables their selection[52]. Values and standard deviations are based on data from three independent experiments. Recipient strains used: wild-type (WT, R1501), *cbpD⁻* (R3966), *cbpD⁻ comM⁻* (R3967), *cbpD⁻ recO⁻ xerS⁻* (R3970), *cbpD⁻ comM⁻ recO⁻ xerS⁻* (R3973)

longer time. Notably, removing ComM from this background suppressed the inflated and elongated phenotype observed in CSP-induced *comX* mutant cells (Supplementary Fig. 4). Our results showed that in *comX* mutant cultures, the phosphorylation state of most of the StkP substrates dropped to barely detectable levels during the 40 min after CSP addition (Fig. 5d and Supplementary Fig. 8e). Analyses of protein levels revealed that the amount of StkP, DivIVA, and another StkP substrate, MapZ, also decreased during this period, but by less than twofold (Supplementary Fig. 8e). Importantly, the same results were obtained for the *comX comM* double mutant (Fig. 5d and Supplementary Fig. 8e). Taken together, these observations indicate that competence development interferes with StkP kinase activity. They further suggest that one or several early *com* genes, with the likely exception of *comM*, are involved in this process. We conclude that induction of at least two parallel pathways lead to the alteration of peptidoglycan synthesis and therefore cell division during early competence development. The ComM pathway inhibits cell constriction by an unknown mechanism; and another pathway requiring another unidentified early *com* gene hinders cell cycle progression by restricting the phosphorylation of StkP substrates, among them DivIVA.

**Biological role of the cell division delay during transformation**. Our data indicate that competent cells undergo cell division after delays in the invagination and closure steps, and that the early competence protein ComM is responsible for this phenomenon. We next asked how cells might benefit from inhibition of division during transformation. The simplest hypothesis is that it preserves genome integrity by allowing cells enough time to complete transformation. The final steps in establishing transformants include integration of transforming DNA into the recipient chromosome by homologous recombination, resolution of resulting heteroduplexes by DNA replication, and segregation of the chromosomes before cell division[50]. These steps prolong the period needed to complete daughter chromosome separation and create the risk that a normally programmed constriction will

produce guillotined DNA and anucleate daughter cells. Yet, comparison of the transformation frequencies for a PCR DNA fragment carrying a single point mutation conferring streptomycin resistance revealed no significant difference between wild-type and "fast dividing" *comM⁻* cells (Supplementary Fig. 9), suggesting that the cell division delay observed in competent cells had no impact on the outcome of transformation. We considered that the need for this delay might be stronger for transformation events that proceed through the formation of chromosome dimers, since these structures may require lengthy periods for resolution and segregation. A recent study has shown that chromosome dimers occur at an unexpectedly high frequency during transformation, with more than 30% of the cells producing a dimer when transformed with pneumococcal chromosomal DNA[51]. It is also known that while chromosomal rearrangements, such as duplications and inversions, are generated fairly frequently by transformation[50,52], they necessitate the creation of chromosome dimers as intermediates. These chromosome dimers are the result of the simultaneous integration of two parts of a single donor DNA fragment into the two daughter chromatids of a partially replicated recipient chromosome (Supplementary Fig. 10). Integration at sister loci of two daughter chromosomes causes the formation of a circular dimer chromosome, which is converted back to monomers after resolution (Supplementary Fig. 10, left). Such transformation products may be difficult to score. By contrast, simultaneous integration of a single repeat sequence at two different copies of this sequence carried by two daughter chromatids may confer a selectable phenotype as it generates one chromosome with a tandem duplication and the other with a deletion (Supplementary Fig. 10, right). This mechanism also accounts for the formation of merodiploid strains harboring large tandem chromosomal duplications previously observed in studies of bacterial transformation[52].

To test whether a cell constriction delay facilitates the establishment of chromosomal rearrangements, we adapted a transformation assay that allows direct selection of merodiploids[52] (see Methods section). This assay is based on the use of PCR fragments as transforming DNA, specifically designed to stimulate the formation of chromosomal tandem duplications. To circumvent a potential fratricide effect, we performed this transformation assay using strains containing a mutant CbpD hydrolase, namely CbpD^C75A [20]. Inactivation of CbpD left the frequency of merodiploid formation by transformation unaltered from that of wild type (Fig. 6), indicating that fratricide does not interfere with merodiploid frequency. However, the frequency of successful merodiploid formation was reduced threefold in cells lacking *comM*, indicating that ~65% of the transformed cells failed to complete the transformation process (Fig. 6). It was recently shown that the RecFOR proteins involved in homologous recombination and the site-specific recombinase XerS are required to resolve chromosome dimers in *S. pneumoniae*, and that their absence reduced the efficiency of merodiploid formation during transformation[51]. We found that cells harboring an inactive CbpD together with the *xerS⁻* and *recO⁻* mutations displayed an 18-fold reduction in merodiploid formation (Fig. 6), which is similar to the effect previously observed in *recO⁻ xerS⁻* cells[51]. We then measured the merodiploid frequency in "fast dividing" *comM⁻* cells lacking these recombination proteins. Introduction of the *comM⁻* mutation into the *cbpD xerS recO* triple mutant resulted in a further twofold drop in merodiploid frequency (i.e., only 3% of transformed cells succeeded in completing the process), suggesting that the failure to slow the cell constriction process sensitizes cells to disruption of dimer resolution. Taken together, these results support a model in which ComM facilitates survival of cells undergoing the resolution of chromosome dimers generated

during transformation by postponing cell division, thereby allowing more time for completion of the process and optimizing the efficiency of transformation.

## Discussion

Whole-genome sequencing indicates that the great plasticity of *S. pneumoniae* results from extensive recombination events that presumably occur through transformation[53,54]. In addition, the high recombination rate occurring during pneumococcal transformation was found to increase the frequency of chromosomal rearrangements that necessitate the generation of chromosome dimers as intermediates in the process[51]. In cells unable to resolve chromosome dimers, or more generally to complete chromosome segregation, cell division resumes following shearing of DNA entrapped by the closing septum, thereby yielding at least one non-viable cell[55]. To avoid this ruinous scenario, bacterial cells have developed overlapping mechanisms that ensure appropriate timing of cell division[56,57]. Our results demonstrate that the ComM-dependent constriction delay observed in competent pneumococcal cells is a crucial element in the completion of transformation events requiring the resolution of chromosome dimer intermediates (Fig. 7). Stable insertion of transforming DNA is a multistep process that may take longer than the doubling time of the cells growing in exponential phase. It is therefore very likely that the cell division delay is specifically programmed during the development of competence to accommodate transformation within the pneumococcal cell cycle.

The mechanism by which ComM regulates peptidoglycan synthesis and cell constriction is unknown. ComM, which is predicted to be an integral membrane protein that is almost fully embedded with 6 or 7 transmembrane segments and short extra-membrane loops, could impair the function of the cell division machinery by directly interacting with one or several of its membrane components. This would be reminiscent of the mechanisms of transient cell division blockage occurring in *Caulobacter crescentus* following DNA damage. In this bacterium, two cell division inhibitors, the membrane proteins SidA and DidA, are induced by DNA damage. Both inhibit cell division by directly binding to proteins of the FtsW/FtsN/FtsI subcomplex, which plays a critical role in cell constriction[9,58]. Another possibility prompted by our finding that the ComM-dependent cell division delay serves to accomodate chromosome dimer resolution would be that this protein is linked to the DNA translocase FtsK, a multifunctional cell division protein[59], best known for stimulating chromosome unlinking and DNA segregation[60]. Interestingly, it has been proposed that FtsK could act as a checkpoint, linked to the septal peptidoglycan synthesis machinery and allowing cell division to proceed once the chromosome has been properly cleared from the closing septum[59,61,62]. Nevertheless, these scenarios are not mutually exclusive and more elaborate ones may exist.

Competence development impinges on the activity of the StkP kinase, the central regulator of pneumococcal cell division. StkP, together with its corresponding phosphatase PhpP and the two paralogs GpsB and DivIVA, plays a major role in cell division and morphogenesis by regulating septal and/or peripheral peptidoglycan synthesis[43,49]. Accordingly, it was proposed that cell elongation is stimulated by non-phosphorylated DivIVA and that DivIVA phosphorylation favors synthesis of the cross wall. *stkP* and *phpP* deletion mutants both exhibit reduced transformation efficiencies[46–48]. This effect could be indirect since peptidoglycan synthesis is impaired and these mutants display severely altered morphology. However, our data showing that expression of early *com* gene(s) perturbs the cell cycle by blocking the phosphorylation of DivIVA and other StkP substrates, are most consistent

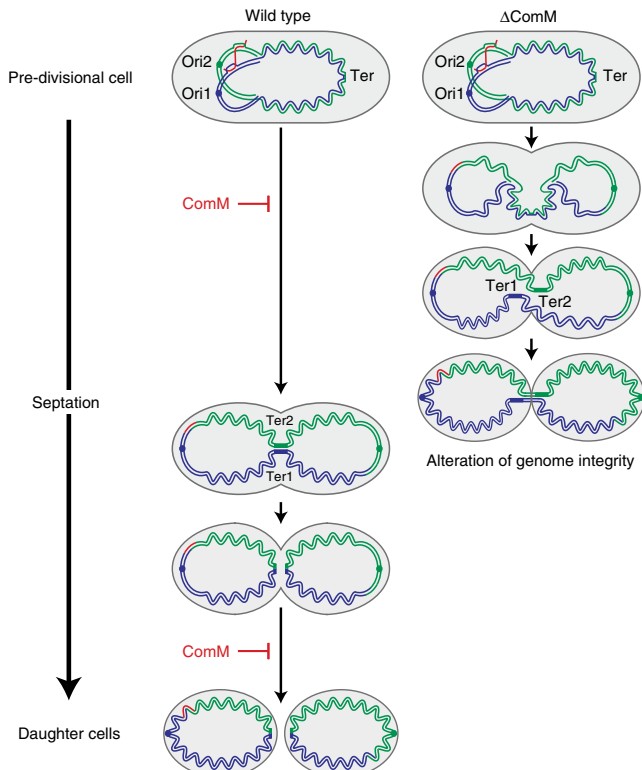

**Fig. 7** ComM plays a role in the resolution of chromosome dimers generated during transformation by postponing cell division. In wild-type cells, ComM delays the formation of daughter cells by inhibiting initiation of septum constriction in pre-divisional cells as well as its closure. This delay allows time for completion of transformation, replication, and chromosome segregation. It benefits transformation events leading to genome rearrangements with the creation of a chromosome dimer as an intermediate. Resolution of chromosome dimers occurs when the termini of the future sister chromosomes are aligned in the septal region[55]. In the absence of ComM, progressive septum closure could entrap DNA resulting in loss of genome integrity. A donor transforming DNA (red) pairing with the two copies of a partially replicated recipient chromosome is shown. The origin (Ori1 and Ori2) and the terminus (Ter1 and Ter2) regions of the sister chromosomes are indicated. For simplicity, only one replication round is shown

with the existence of a functional link between StkP signaling and competence.

Several lines of evidence indicate that competence is tightly and specifically integrated into the life style of each bacterial species[63]. Among other transformable bacteria, only *B. subtilis* has been shown to undergo a prolonged growth arrest when competent[64–67]. In this species, competence, also called the K-state, is a bistable state that develops in a minority of cells at the onset of the stationary phase. When fresh medium is added to the culture and cells resume growth, the competent cells remain in a kind of dormant state, and it takes several hours before they start to grow again[64,66,67]. This arrest is more accurately referred to as "outgrowth delay" since cells are not actively growing when it occurs. At least three factors induced during competence are collectively required to generate this phenotype: ComGA restrains the rate of cell growth by sequestering both the MreB protein involved in the regulation of cell elongation[67] and the RelA protein responsible for activation of the stringent response, a highly conserved response to nutrient starvation stress[66]; Maf redundantly inhibits cell division in cells lacking ComGA[65], and another unidentified protein contributes to RNA synthesis

inhibition[66]. As a fraction of competent cells exhibits tolerance to antibiotics, it is proposed that the K-state represents a form of persistence. Thus, the bistable expression of both transformability and growth arrest could have evolved in *B. subtilis* as an adaptation for survival in a variable environment, such as soil[66]. In contrast to the situation in *B. subtilis*, it is remarkable that competence arises in the whole pneumococcal population for a short period of time while cells undergo active multiplication (note that there is no homolog of *maf* in *S. pneumoniae* and that the constriction delay is maintained in the absence of ComGA in competent pneumococcal cells, see Supplementary Fig. 11). *S. pneumoniae* is a human commensal that usually inhabits the nasopharynx and can cause several serious invasive diseases, including pneumonia, otitis, sinusitis, meningitis, and septicemia[68]. Genetic diversity is a key element in the pathogenicity of the pneumococcus. Our data suggest that the ComM-dependent cell division delay provides this species with maximal adaptation potential by optimizing the transformation process without compromising cell viability. Finally, as the crucial role of Ser/Thr kinases in the bacterial cell cycle is now well established[69], the discovery of inhibitors that target their pathways may provide new strategies for antibiotic development.

## Methods

**General methods.** *S. pneumoniae* strains (Supplementary Table 1) were all derived from strain R800[70], except for the serotype 19F encapsulated strain G54[31]. Stock cultures of pneumococal strains were routinely grown at 37 °C in Todd–Hewitt (BD Diagnostic System) plus yeast extract (THY) medium to OD$_{550}$ 0.3; after addition of 15% (vol/vol) glycerol, stocks were kept frozen at −70 °C. These precultures were used to initiate cultures in C + Y medium at $6 \times 10^6$ cells per ml. The C + Y medium, derived from the C medium described by Tomasz in 1967[71], contained the following ingredients per liter: 5 g casein hydrolysate, 6 mg tryptophane, 11.25 mg cystine, 2 g sodium acetate, 8.5 g K$_2$HPO$_4$, 0.5 g MgCl$_2$·6H$_2$0, 12.5 mg CaCl$_2$, 250 μg MnCl$_2$, 0.5 mg FeSO$_4$·7H$_2$O, 0.5 mg CuSO$_4$·5H$_2$O, 0.5 mg ZnSO$_4$·7H$_2$O, 0.6 μg biotin, 0.6 mg nicotinic acid, 0.7 mg pyridoxine-HCl, 0.6 mg thiamine-HCl, 0.3 mg riboflavin, 2.4 mg calcium pantothenate, 50 mg L-asparagine·H$_2$O, 20 mg uridine, 20 mg adenosine, 22 mg glutamine, 0.3 g sodium pyruvate, 0.3 g saccharose, 2 g glucose, 0.8 g bovine serum albumin, 5 mg choline, and 25 g yeast extract. Transformation was performed as described[24]. Transformation efficiency with PCR DNA fragments was performed as described[72] using a 4.2-kb fragment carrying the *rpsL41* point mutation conferring resistance to streptomycin (Strep). Antibiotic concentrations (μg ml$^{-1}$) used for the selection of *S. pneumoniae* transformants were: chloramphenicol (Cat), 4.5; erythromycin (Ery), 0.05; kanamycin (Kan), 250; rifampicin (Rif), 2; spectinomycin (Spec), 100; streptomycin, 200; tetracycline (Tet), 1. For the monitoring of growth and *luc* expression, aliquots from stock cultures were first inoculated at OD$_{550}$ 0.006 in C + Y medium supplemented with 20 mM HCl. Cells were grown to OD$_{550}$ 0.1; then diluted to OD$_{550}$ 0.01 in luciferin containing C + Y medium supplemented with 8 mM HCl[73], and subsequently distributed (300 μl per well) into a 96-well white microplate with clear bottom. The cultures were grown to OD$_{550}$ 0.06 and synthetic CSP1 (25 ng ml$^{-1}$) was added. Relative luminescence unit and OD values were recorded throughout incubation at 37 °C in a Varioskan Flash luminometer (Thermo 399 Electron Corporation). These experiments were made in triplicates.

Note that for all experiments, C + Y medium was supplemented with HCl (10 mM final concentration unless otherwise indicated) to prevent spontaneous development of competence except for strains harboring the Δ*comC* mutation[17], which cannot develop competence spontaneously.

Details of plasmid and strain constructions are described in Supplementary Methods. Strains, plasmids, and oligonucleotide primers used for PCR, RTqPCR, and site-directed mutagenesis are listed in Supplementary Tables 1–3.

**Reverse transcription quantitative real-time PCR (RTqPCR).** Three milliliter cultures grown in C + Y medium at 37 °C at OD$_{550}$ of 0.1 were treated, or not, with CSP for 5 min, and diluted two times in C + Y medium maintained at 4 °C. Total RNA was extracted using the RNeasy Mini kit (Qiagen) and treated with DNase using TURBO DNA-free™ (Ambion®) according to the manufacturer's instructions. The reverse transcription reaction and the quantitative PCR were performed using the iTaq™ Universal SYBR® Green One-Step Kit (Bio-Rad). RNA samples were diluted 1:30 and 5 μl was added to the PCR reaction mixture (final volume 20 μl) as recommended by the manufacturer. Oligonucleotides OCN186 and OCN187 were selected for expression analysis of the *lytR* gene. Oligonucleotides specific for *rpoB* were used as control as expression of *rpoB* does not change during competence induction[15,17].

RTqPCR was carried out with the Light Cycler® 480 II system (Roche). The run protocol consisted of a reverse transcription reaction at 50 °C for 10 min, and a

quantitative real-time PCR with a preincubation step at 95 °C for 5 min, 45 cycles of amplification at 95 °C for 10 s, 50 °C for 10 s, 72 °C for 10 s; a melting curve and a final cooling at 37 °C for 1 min. Specific amplifications were confirmed by single peaks in melting curve analysis. Cycle threshold (CT) values were obtained according to the software instructions. Relative quantification was performed with the 2-ΔΔC T method. Each PCR reaction, run in duplicate for each sample, was repeated for at least two independent times. Data are represented as mean ± s.e.m. Statistical analysis was conducted using Graph Pad Prism 5.0 (San Diego, CA), *F*-tests indicated that the variance between groups was not significantly different (*P* = 0.563), and statistical differences were determined using a two-tailed unpaired *t*-test.

**Merodiploid formation by transformation.** Merodiploid formation was investigated by transformation as previously described[52], with modifications. It involves transformation with a short repeat sequence (hereafter called R$_1$ or R$_2$) adjacent to a non-repeat flanking sequence (hereafter called A or Z). R$_1$-A and R$_2$-Z are both present in the pneumococcal chromosome. Transformation with either or both of these fragments generates a merodiploid chromosome with a tandem duplication of 107.4 kb containing the essential gene *codY*[74]. Selection of merodiploids can therefore be achieved by co-transformation of R$_1$-A and R$_2$-Z with a *codY::trim* cassette inactivating *codY*. Since *codY* is essential, only merodipoids with two copies of *codY* will be able to viably accept this cassette, and all recovered trimethoprim-resistant clones will thus be merodiploids, allowing comparison of frequency of merodiploid formation in different genetic contexts. CSP-induced transformation was performed in C + Y medium, using pre-competent cells treated at 37 °C for 8 min with synthetic CSP1 (25 ng ml$^{-1}$). An aliquot of 100 μl of this culture was added to a mix of transforming DNA containing 10 ng of *codY::trim* cassette[74] generated by PCR (codY7/codY8 primers and TD80 DNA as template, 7371 bp), 5 ng of R$_1$-A PCR (merod-a1/merod-b primers and R800 DNA as template, 2980 bp) and 5 ng of R$_2$-Z PCR (merod-a2/merod-c primers and R800 DNA as template, 2793 bp). Cells and transforming DNA were incubated for 20 min at 30 °C. An aliquot of 200 μl of pre-warmed C + Y medium was added and the culture was further incubated for 30 min at 30 °C. Transformants were selected by plating on CAT-agar supplemented with 4% (vol/vol) horse blood, followed by selection using a 10 ml overlay containing trimethoprim (20 μg ml$^{-1}$), after phenotypic expression for 90 min at 37 °C.

**Fluorescence microscopy and analysis.** Pneumococcal precultures grown in C + Y medium at 37 °C to an OD$_{550}$ of 0.1 were induced to develop competence. At indicated times after CSP addition, 1 ml samples were collected, cooled down by addition of 500 μl of medium pre-cooled at 4 °C, pelleted (3 min, 3000×g) and resuspended in 50 μl C + Y medium. An aliquot of 2 μl of this suspension was spotted on a microscope slide containing a slab of 1.2% C + Y agarose before imaging, as described[75]. Images were captured and processed using the Nis-Elements AR software (Nikon).

To classify and quantify pneumococcal cells in different stages of the cell cycle, phase contrast images were further analyzed using the Metamorph 7.5 software and the different cell categories (Fig. 1a) were detected with the integrated morphometric analysis tool. Single cells were first detected using the threshold command from Metamorph. We then wrote a journal to automatically measure morphological parameters including the cell area, the cell perimeter, the length and the width, the elliptical factor, and the shape factor (see Supplementary Table 4). The elliptical factor is the ratio of the cell's width to its length and increases with the cell cycle. The shape factor is calculated from the perimeter (*P*) and the area (*A*), SF = $4\pi A/P^2$. It allows distinction between round newborn cells and dividing longer cells with a midcell constriction. Indeed, values of the shape factor close to 0 represent elongated cells, whereas a value of 1 corresponds to a perfect circle. The value of the shape factor also decreases with the constriction degree of dividing cells. Measurements of the cell length and the outer radius, the maximal distance from the center to the contour of the cell, were further used to refine categories. Using these parameters, cells were then automatically classified into two groups, each containing three different categories as shown in Fig. 1a. Group I corresponds to elongating cells (without a constriction), while cells in group II are dividing cells exhibiting a midcell constriction. In Group I, cells of category A are newborn cells, cells of category C are pre-divisional cells and cells of category B are at an intermediate stage between cells of categories A and C (Fig. 1a). In group II, cells of category D, E, and F are distinguished by the degree of constriction, with cells of category D initiating the constriction process, and cells of category F finishing it (Fig. 1a and Supplementary Fig. 2). Analyses of technical replicates indicated that results were reproducible for images containing about 200 cells. Note that a lower-phase contrast intensity at the point of contact between two newborn cells allowed distinction between two cells of category A (newborn cells) and one single cell of category F (future daughter cells still connected by membrane and cytoplasm). To test the reliability of this method, we measured GFP intensity profiles along the long axis of the cells (Supplementary Fig. 2). A pair of non-dividing cells exhibited two well-separated fluorescence peaks (see cells labeled 6, 4, 19, and 38 in Supplementary Fig. 2h). In contrast, as the cytoplasms of the two future daughter cells of cells of category F are still connected, the fluorescence intensity in the central dip in the plot remained greater than half of the maximal intensity (see cells labeled 26, 28, 33, 34, and 42 in Supplementary Fig. 2h). Values for parameters

used to classify cells in the different categories are indicated in Supplementary Table 4. The same values were used to analyze phase contrast images of the G54 encapsulated strain and classify the cells in the different stages of the cell cycle. The proportion of dividing cells in these images is probably overestimated as the distinction between cells of categories F and newborn cells was not as clear as for non-capsulated strains. Violin plots representing the cell size distribution in each cell category were generated using ggplot2[76] and the software R 3.4.1[77].

**Peptidoglycan labeling with fluorescent D-amino acids.** Cultures grown in C + Y medium at 37 °C to an $OD_{550}$ of 0.1 were separated into two portions. One portion was treated with CSP and the control cultures continued to grow without CSP. Ten minute after CSP addition, cells were incubated for another 15 min at 37 °C with 70 mM of HADA (a fluorescent hydroxy coumarin derivative of D-alanine)[41] and washed three times with cold C + Y medium by centrifugation at 4 °C. Cells were spotted on a microscope slide containing C + Y medium as described above and allowed to grow 10–15 min at room temperature before imaging.

**Immunoblot analysis.** Cultures grown in C + Y medium at 37 °C to an $OD_{550}$ of 0.1 were separated into two portions. One portion was treated with CSP for 10 min (strains in $comX^+$ background: R1501, R3966, and R3967) or 40 min (strains in $comX^-$ background: R2002 and R2132) and the control cultures continued to grow without CSP. Samples were then cooled rapidly by transferring the cultures in glass tubes containing pre-cooled C + Y medium and maintained below 4 °C in a bath containing a mixture of ice and NaCl. The cell pellets were resuspended in pre-cooled TE-buffer (10 mM Tris, 1 mM EDTA, pH 7.5) supplemented with Complete antiprotease (Roche Diagnostics) and anti-phosphatase cocktails II and III (Sigma-Aldrich), before disruption by bead-beating with glass beads (425–600 µm, Sigma-Aldrich) at 5800 rpm two times for 30 s and two times for 20 s with 2 min intervals (Precellys 24 lysis and homogenization, Bertin Technologies). The protein concentration was determined using the Bradford protein assay (Bio-Rad) and the samples were further diluted in 5× SDS-sample buffer. An aliquot of 20 µg of each crude extract was separated on an 8% SDS polyacrylamide gel electrophoresis after boiling for 5 min, and transferred to a PVDF membrane by western blotting. In vivo phosphorylated proteins were immunodetected using an anti-phosphothreonine polyclonal antibody (Cell Signaling, #9381) at 1:2000 as described previously[44]. StkP was detected using a rabbit polyclonal antibody specific for the StkP extracellular domain (anti-StkP-PASTA) diluted at 1/175,000[44]. To detect MapZ and DivIVA, purified MapZ (cytoplasmic domain) and DivIVA were used to immunize rabbits (Eurogentec) for the production of polyclonal antibodies that were diluted at 1:5000 in western blot experiments.

Quantification of western blot signals for StkP, MapZ, and DivIVA was performed as follows. The same volume of protein from the same samples analyzed with the anti-phosphothreonine antibody was loaded on Biorad mini-PROTEAN TGX stain-free 4–15% pre-cast gel. Gels were activated by ultraviolet (UV) exposure for 45 s using a Bio-Rad ChemiDoc MP imager to visualize and estimate total protein per lane. Proteins were then transferred to a PVDF membrane, using a Turbo Blot transfer unit (Bio-Rad). After incubation with primary and secondary antibodies, membranes were imaged for the total protein transferred using the stain-free application on the ChemiDoc MP imager (Bio-Rad). Membranes were further incubated with clarity chemiluminescence substrate (Bio-Rad), and imaged on the ChemiDoc MP. Detection and measurement of band intensities was performed using Image Lab 5.2 software (Bio-Rad), automatically normalized with respect to total protein quantification. Statistical analysis was conducted using Graph Pad Prism 5.0 (San Diego, CA), $F$-tests indicated that the variance between groups was not significantly different ($P = 0.439$), and statistical differences were determined using a two-way anova test followed by Bonferroni's post-test.

**Data availability.** All relevant data supporting the findings of the study are available in this article and its Supplementary Information files, or from the corresponding authors on request.

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

## Acknowledgements

We are indebted to Jean-Pierre Claverys who first anticipated a link between competence and the cell cycle. We thank Bernard Martin and Chantal Granadel for constructing the *comM* Janus mutant, and Aurore Fleurie for help in generating the phosphorylation profiles. We thank Mathieu A. Bergé, Jean-Yves Bouet, Calum Johnston, and Dave Lane for stimulating discussions and critical reading of the manuscript as well as other members of the Polard laboratory, Jérôme Rech, and David Rudner for helpful comments and support. This work was funded by the Centre National de la Recherche Scientifique, Université Paul Sabatier, Agence Nationale de la Recherche (Grant ANR-10-BLAN-1331). M.J.B. was supported by a PhD grant from Ministère de la Recherche et de l'Enseignement. M.S.V. was supported by a National Institutes of Health grant (GM113172). Funding for the video microscopy equipment was provided by the Fonds Européen de Développement Régional Midi-Pyrénées and the Association pour la Recherche sur le Cancer.

## Author contributions

M.J.B., P.P. and N.C. planned the experiments, analyzed and interpreted the data. M.J.B. and N.C. performed all experiments except for the construction of the *comM* janus. N.C. carried out the FDAA staining with the supervision of Y.V.B. N.C. performed image analyses. C.M., I.M.-B., C.G. and N.C. generated and interpreted the anti-

phosphothreonine western blot analysis. Y.V.B. and M.S.V. provided new reagents. C.G., Y.V.B., C.M., I.M.-B. and M.S.V. commented on the manuscript. P.P. and N.C. wrote the manuscript with contributions from M.J.B.

## Additional information

**Competing interests:** The authors declare no competing financial interests.

