## [Peer Review File · Nature Communications]

Editorial Note: this manuscript has been previously reviewed at another journal that is not operating a transparent peer review scheme. This document only contains reviewer comments and rebuttal letters for versions considered at Nature Communications. Parts of this peer review file have been redacted.

REVIEWERS' COMMENTS:

Reviewer #2 (Remarks to the Author):

I wish to commend the authors for the additional experimentation and thoroughness in their responses to all the reviewers' comments. My primary concern from the first draft was the unconvincing data presented for phosphorylation of DivIVA. I am pleased to see not only additional evidence is now provided to clarify the amount of DivIVA phosphorylated, but that a second protein, Jag, is an identified target. These findings further substantiate and clarify the model for ComM dependence in division inhibition. Overall, I am satisfied with the revisions and have no further significant concerns. A minor suggestion, which is more a semantic issue, is referring to CSP-stimulated cells as "competent cells" throughout the manuscript; this only becomes a (minor) issue in comX mutants, which would not be able to become "competent". I understand that for consistency in referring to "competence" throughout the manuscript that this is a reasonable use of the term. "CSP-induced cells" could be an alternative way to refer to "competent cells".

Reviewer #3 (Remarks to the Author):

The manuscript by Berge and colleagues has been significantly improved. Most importantly the authors have now showed that the effect they observe on DivIVA phosphorylation state, although small, is reproducible. Furthermore they show that ComM is not involved in StkP kinase activity. The authors have adequately addressed all my concerns and therefore I have not further comments except the two minor ones below:

Line 277 – do authors mean "peripheral and septal syntheses" instead of "peripheral and peptidoglycan syntheses"?

Line 283 – for how long were cells allowed to grow on the slide? From reply to reviewer 1, question 2.2 b, I assume this was not a controlled time interval. If that is the case, this should be mentioned in the main text. Also, that may cause avoidable variability in the data.

Reviewer #4 (Remarks to the Author):

This is a very exciting study that shows clear connections between the cell cycle and pneumococcal competence. Specifically, it is shown that ComM, which is under control of ComE, induces a block in cell division. A role for StkP in competence dependent block of cell division is also revealed. I have reviewed this paper before for [redacted] and the authors have very thoroughly addressed my comments (JW Veening, UNIL). Specifically, the StkP gels look much more convincing and, even though it hasn't revealed the mechanisms at play yet, this work now opens much work and there are exciting times ahead for the competence field.

Comments:

- 1) L23, exogenous DNA can be actively imported? (if there is no DNA around, it will not be imported)
- 2) L33. Are you sure about using the word 'programmed'? This would indicate an intricate level of

regulation.

'A regulated cell division....' would also work very nicely as title. comM transcription is activated by ComE.

I leave this up to the authors.

3) L89, why not also cite reference 70 (Martin et al., MolMicro 2010) (and our work, Slager et al. 2014, if you like) here that demonstrates this, which is cited later in the supplementary references?

4) L466, replication instead of multiplication?

5) L474, in the bacterial cell cycle

6) L484, This might be a good opportunity to provide your recipe for C+Y (I get frequently asked for it and cannot provide a good reference; the most extensive, but not completely accurate paper I could find describing C+Y is Adams and Roe, Jbact 1995).

7) L554. Reference 73 writes: '2 μ L of this suspension were spotted on a microscope slide containing a slab of 1.2% C+Y agarose as described previously [48] before imaging.' It would be more correct to cite the de Jong et al study here. I know it's a bit awkward to ask you to cite one of my papers, but otherwise a paper trail (like for the C+Y medium) is generated.

Reviewer #2 (Remarks to the Author):

I wish to commend the authors for the additional experimentation and thoroughness in their responses to all the reviewers' comments. My primary concern from the first draft was the unconvincing data presented for phosphorylation of DivIVA. I am pleased to see not only additional evidence is now provided to clarify the amount of DivIVA phosphorylated, but that a second protein, Jag, is an identified target. These findings further substantiate and clarify the model for ComM dependence in division inhibition. Overall, I am satisfied with the revisions and have no further significant concerns. A minor suggestion, which is more a semantic issue, is referring to CSP-stimulated cells as "competent cells" throughout the manuscript; this only becomes a (minor) issue in *comX* mutants, which would not be able to become "competent". I understand that for consistency in referring to "competence" throughout the manuscript that this is a reasonable use of the term. "CSP-induced cells" could be an alternative way to refer to "competent cells".

We understand the point made by this reviewer. As suggested, we replaced "competent cells" by "CSP-induced *comX* mutant cells" in the paragraphs and the figures presenting the results related to the *comX* mutant.

Reviewer #3 (Remarks to the Author):

The manuscript by Berge and colleagues has been significantly improved. Most importantly the authors have now showed that the effect they observe on DivIVA phosphorylation state, although small, is reproducible. Furthermore they show that ComM is not involved in StkP kinase activity.

The authors have adequately addressed all my concerns and therefore I have not further comments except the two minor ones below:

Line 277 – do authors mean "peripheral and septal syntheses" instead of "peripheral and peptidoglycan syntheses"?

We thank this reviewer for noticing this error. We modified the text with: "peripheral and septal peptidoglycan syntheses".

Line 283 – for how long were cells allowed to grow on the slide? From reply to reviewer 1, question 2.2 b, I assume this was not a controlled time interval. If that is the case, this should be mentioned in the main text. Also, that may cause avoidable variability in the data.

We didn't rigorously control the time interval after the cells were spotted on the microscope slide and before imaging. However, we realized that the time to prepare samples influenced our results. We thus kept it to a minimum by cultivating and treating the cells consecutively. We estimate the time the cells were allowed to grow on the microscope slide before imaging to be about 10 to 15 minutes. We now clarified this point in the main text and we also added a sentence in the Methods section.

Reviewer #4 (Remarks to the Author):

This is a very exciting study that shows clear connections between the cell cycle and pneumococcal competence. Specifically, it is shown that ComM, which is under control of ComE, induces a block in cell division. A role for StkP in competence dependent block of cell division is also revealed. I have reviewed this paper before for [redacted] and the authors have very thoroughly addressed my comments (JW Veening, UNIL). Specifically, the StkP gels look much more convincing and, even though it hasn't revealed the mechanisms at play yet, this work now opens much work and there are exciting times ahead for the competence field.

Comments:

1) L23, exogenous DNA can be actively imported? (if there is no DNA around, it will not be imported)

This sentence has been modified to shorten the abstract as follows: "Competence for genetic transformation is a differentiation program during which exogenous DNA is imported into the cell and integrated into the chromosome". When exogenous DNA is present, it is imported in competent cells. In fact, unless DNase is added to the medium, exogenous DNA is always present. This is true in laboratory cultures (even during exponential phase), and this is particularly true in the natural environment of the pneumococcus.

2) L33. Are you sure about using the word 'programmed'? This would indicate an intricate level of regulation.

'A regulated cell division....' would also work very nicely as title. comM transcription is activated by ComE.

I leave this up to the authors.

We would prefer to keep the word "programmed". Here, we don't use this word to refer to the genetic regulation of the cell division arrest. We mean to say that this arrest is planned during the development of competence for a specific purpose. Our results suggest that the cell division arrest ensures that transformation is complete before resumption of cell division. It may preserve physical chromosome integrity during transformation and thus provides *S. pneumoniae* with the maximum potential for genetic diversity and adaptation.

3) L89, why not also cite reference 70 (Martin et al., MolMicro 2010) (and our work, Slager et al. 2014, if you like) here that demonstrates this, which is cited later in the supplementary references?

We agree and we introduced these references as suggested.

4) L466, replication instead of multiplication?

Here, we use the term "active multiplication" in the sense of "active cell cycle". The point we want to make is that it is quite remarkable that competence arises in the whole pneumococcal population for a short period of time during the exponential growth phase. During this phase, most of the cells are indeed actively replicating their genetic material but they also undergo active transcription and translation, they are doubling their cell mass and dividing.

5) L474, in the bacterial cell cycle

We modified the text as suggested.

6) L484, This might be a good opportunity to provide your recipe for C+Y (I get frequently asked for it and cannot provide a good reference; the most extensive, but not completely accurate paper I could find describing C+Y is Adams and Roe, Jbact 1995).

We agree. We now provide the composition of the C+Y medium routinely used in the team in the Methods section.

7) L554. Reference 73 writes: '2 μ L of this suspension were spotted on a microscope slide containing a slab of 1.2% C+Y agarose as described previously [48] before imaging.' It would be more correct to cite the de Jong et al study here. I know it's a bit awkward to ask you to cite one of my papers, but otherwise a paper trail (like for the C+Y medium) is generated.

We agree and we corrected this reference.